# Genome-Wide Identification and Characterization of the Msr Gene Family in Alfalfa under Abiotic Stress

**DOI:** 10.3390/ijms24119638

**Published:** 2023-06-01

**Authors:** Xianglong Zhao, Xiao Han, Xuran Lu, Haoyue Yang, Zeng-Yu Wang, Maofeng Chai

**Affiliations:** Key Laboratory of National Forestry and Grassland Administration on Grassland Resources and Ecology in the Yellow River Delta, College of Grassland Science, Qingdao Agricultural University, Qingdao 266109, China; xlzhao23@163.com (X.Z.); 18266391263@163.com (X.H.); 13851390970@163.com (X.L.); haoyueyang2021@163.com (H.Y.); zywang@qau.ac.cn (Z.-Y.W.)

**Keywords:** alfalfa, gene family, *Msr* genes, abiotic stress, expression profile

## Abstract

Alfalfa (*Medicago sativa*) is an important leguminous forage, known as the “The Queen of Forages”. Abiotic stress seriously limits the growth and development of alfalfa, and improving the yield and quality has become an important research area. However, little is known about the Msr (methionine sulfoxide reductase) gene family in alfalfa. In this study, 15 *Msr* genes were identified through examining the genome of the alfalfa “Xinjiang DaYe”. The MsMsr genes differ in gene structure and conserved protein motifs. Many cis-acting regulatory elements related to the stress response were found in the promoter regions of these genes. In addition, a transcriptional analysis and qRT-PCR (quantitative reverse transcription PCR) showed that *MsMsr* genes show expression changes in response to abiotic stress in various tissues. Overall, our results suggest that *MsMsr* genes play an important role in the response to abiotic stress for alfalfa.

## 1. Introduction

Alfalfa (*Medicago sativa*) is a nutrient-dense forage crop, providing an abundance of crude protein, carbohydrates, vitamins B, C, and E, as well as various other micronutrients [1]. With an estimated global planting area of 30 million hectares, alfalfa is the world’s second-largest leguminous forage crop after soybean (*Glycine max*) and is widely cultivated in the Xinjiang, Gansu, and Qinghai provinces of China [2]. However, the degradation of the ecological environment has limited the efficiency and quality of alfalfa production. Fortunately, “Xinjiang Daye” is a homotetraploid cultivar offering a high-quality chromosome-level assembly with 32 chromosomes, providing an invaluable resource for selecting key stress-related genes for genetic engineering and enhancing alfalfa’s stress resistance [3].

Methionine (Met) is one of the most easily oxidized amino acids. Under oxidative stress, Met is quickly converted to methionine sulfoxide (MetO), which can be reduced back to Met via methionine sulfoxide reductase (Msr). This process plays an important role in protecting cells from oxidative damage. [4,5]. Excessive reactive oxygen species (ROS) in plants causes oxidative damage to proteins, polysaccharides, lipids, DNA, and RNA, and can even cause plant death [6]. Upon oxidation, methionine is converted into two diastereomers, methionine-S-sulfoxide (Met-S-SO) and methionine-R-sulfoxide (Met-R-SO). Msr contains two subfamilies, MsrA (methionine sulfoxide reductase A) and MsrB (methionine sulfoxide reductase B). MsrA reduces Met-S-SO, while MsrB reduces Met-R-SO, thereby balancing the content of both oxidized forms in response to oxidative stress [7,8]. In particular, MsrA is present in all known eukaryotes and prokaryotes as well as most archaea, while MsrB is present in all eukaryotes [7,9,10]. Studies have demonstrated that Msr proteins play a crucial role in the response to oxidative stress. In yeast, MsrA confers resistance to H_2_O_2_ [11,12]. In mammalian cells, the expression of *Msr* decreases in response to aging and disease [13,14]. *MsrA* was the first *Msr* to be isolated [15]. Only one *MsrA* gene has been detected in animals, while plants have multiple *MsrA* genes. The member number and gene function of *Msr* gene family have been reported in several plants. The *Arabidopsis thaliana* genome contains five *MsrAs* and nine *MsrBs* [16]. Rice contains four *MsrAs* and three *MsrBs* [17]. Soybean contains seven *MsrAs* and five *MsrBs*, some of which have been characterized [18]. There are three *MsrAs* and three *MsrBs* in maize (*Zea mays*) [19]. Four *MsrAs* and eight *MsrBs* are located in the wheat genome, and relatively comprehensive functional assays of *TaMsrA4*, *TaMsrB3.1,* and *TaMsrB5.2* have been carried out [20,21,22].

*Msrs* are multifunctional, and their functions have been partially verified in the model plant Arabidopsis, such as protecting against oxidative stress from sources including methyl viologen, ozone, and high light intensity [23]. An *MsrA2* Arabidopsis mutant showed slow growth [24]. Additionally, *MSRBs* play an important role in enabling tolerance to oxidative damage and in preserving photosynthesis antennae, a vital factor for sustaining vegetative growth under environmental constraints [25,26]. Overexpression of both *MsrB7* and *MsrB8* in Arabidopsis was shown to increase tolerance to the herbicide methyl viridine [27]. Overexpression of MsrB1 and MsrB2 was found to enhance seed longevity in Arabidopsis [28]. However, relatively little research has been carried out on the *Msr* gene family in forage crops. In wheat *(Triticum aestivum*), *TaMsrA4.1* increased seedling tolerance to salt and drought, and *TaMsrA3.1* increased seedling tolerance to osmotic pressure. In soybean (*Glycine soja*), the interaction of *GsCBRLK* and *GsMsrB5a* improved the tolerance of plants to salt and alkali. Allogeneic expression of *ZmMsrB1* in maize enhanced Arabidopsis salt stress tolerance [29]. *OsMsrB5* increased seed vigor and longevity in rice (*Oryza sativa*) [30]. In addition to that, there have been increasing reports on *Msr* genes in fruits and vegetables in recent years. In particular, in bananas (*Musa acuminata*), the redox regulation of the transcription factor *MaNAC42* mediated by *MaMsrB2* may be involved in the regulation of banana fruit ripening by controlling the expression of its target genes [31]. *MaMsrA4* mediates the redox regulation of the ethylene signaling component *MaEIL9* to regulate banana fruit ripening [32]. In tomato (*Solanum lycopersicum*), *SlMsrB2* plays a role in drought tolerance and promotes chlorophyll accumulation by regulating ROS accumulation [33]. In kiwifruit (*Actinidia deliciosa*), *AdMsrB1* plays a role in ethylene synthesis and is involved in the ripening process [34]. In summary, the Msr gene family has a rich biological function. Therefore, the Msr gene family has great research potential and value.

Currently, similar bioinformatics analyses have been carried out on the Msr gene family in *Brachypodium distachyon* and soybean, which reveal the possible functions of the Msr gene family in these two species [18,35]. In this study, we used the genomic data of “Xinjiang DaYe” alfalfa as a foundation to systematically investigate the Msr gene family in alfalfa. In addition to basic physicochemical and sequence analyses of the Msr gene family, this study conducted a detailed analysis of the gene duplication relationships and cis-regulatory elements of the Msr gene family in alfalfa. In the conserved domain analysis, other conserved domains of gene families were found in the MsMsr gene family, suggesting the possibility of gene fusion. Furthermore, this study analyzed the expression levels of Msr genes in roots, stems, and leaves, and analyzed the expression patterns of the MsMsr gene family under salt, drought, and ABA (abscisic acid) stress using transcriptome and qRT-PCR (real-time quantitative polymerase chain reaction) analyses. Based on the comprehensive analysis of the MsMsr gene family, it was found that *MsMsrA7* and *MsMsrB6* may play a positive role in responding to abiotic stress. This study provides a systematic analysis of the Msr gene family in alfalfa and contributes to the exploration of the functions and molecular breeding of abiotic stress tolerance in this important economic crop.

## 2. Results

### 2.1. Identification of MsMsr Family Members

By comparing and deleting redundant data in the alfalfa genome using *MsrA* and *MsrB* hidden Markov models (HMM), a total of 15 MsMsr family members were identified in the alfalfa genome of “Xinjiang Danye” (Table 1). The length of the coding sequences (CDS) of the 15 *MsMsrs* ranged from 420 bp (*MsMsrB4*) to 2796 bp (*MsMsrB6*). Accordingly, MsMsrB4 had the smallest theoretical molecular weight (15,101.88 Da), and MsMsrB6 had the largest theoretical molecular weight (101,351.14 Da). The isoelectric point (pI) of MsMsrs ranged from 5.2 (MsMsrA9) to 9.09 (MsMsrB2). Nine MsMsrs had isoelectric points lower than seven, and six had isoelectric points greater than seven. In addition, this study also conducted subcellular localization and signal peptide prediction for 15 MsMsr proteins (Appendix A). The signal peptides of 10 MsMsr proteins were analyzed, and 4 (MsMsrA1, MsMsrA4, MsMsrB2, and MsMsr5) were found to contain chloroplast transit peptides, while 2 (MsMsrB1 and MsMsrB3) contained mitochondrial transit peptides, and MsMsrB4 contained a peroxisomal targeting signal. In subcellular localization prediction, five MsMsr proteins (MsMsrA1, MsMsrA4, MsMsrB2, MsMsrB5, and MsMsrB6) were found to be located in the plastids, supporting the presence of chloroplast transit peptides. In addition, four MsMsr proteins (MsMsrA2, MsMsrA3, MsMsrA8, and MsMsrA9) were located in the cytoplasm, and three MsMsr proteins (MsMsrA5, MsMsrA6, and MsMsrA7) were found in the endoplasmic reticulum. MsMsrB1 and MsMsrB3 were located in the cytoplasm and mitochondria, while MsMsrB4 was found in the cytoplasm and peroxisomes.

### 2.2. Phylogenetic Analysis of Msrs

A multi-species evolutionary tree of Msr was constructed using the maximum likelihood (ML) method in MEGA 7.0, which included four species: *M. sativa*, *M. truncatula*, *G. max*, and *A. thaliana*, and their corresponding bootstrap values were listed [36] (Figure 1). Msr proteins form two subfamilies. Subgroup I (MsrA) contains 22 genes, and subgroup II (MsrB) contains 25 genes (Appendix A).

Specifically, in the MsrA subfamily, *MsMsrA5*, *MsMsrA6*, *MsMsrA7*, *MtMsrA4*, *AtMsrA5*, and *GmMsrA5* were collected together in the phylogenetic tree, suggesting a close homology between them. Among them, *MtMsrA4*, *MsMsrA5*, *MsMsrA6*, and *MsMsrA7* have a closer homology relationship, followed by *GmMsrA5*. In addition, *MsMsrA3*, *MsMsrA8*, *MsMsrA9*, *GmMsrA4*, and *MtMsrA3* were grouped in the same clade, and the results showed that *MsMsrA3*, *MsMsrA8*, *MsMsrA9*, and *MtMsrA3* were more closely homologous. This indicates that the homology of alfalfa and *M. truncatula* is higher than that of soybean.

In the MsrB subfamily, the MsrB subfamily has been divided into two branches, in which *GmMsrB1*, *AtMsrB1*, *MtMsrB1*, *MsMsrB3*, *MsMsrB1*, and *MsMsrB2* are classified into the same branch, while genes of other MsrB members are classified into the other branch. Among them, *MsMsrB1* and *MsMsrB2* exhibit higher homology, while *MtMsrB1* and *MsMsrB3* exhibit higher homology. *MsMsrB5* and *MtMsrB4* have a higher homology.

### 2.3. Sequence and Structure Analysis of MsMsrs

A domain analysis clearly showed that *MsMsrAs* had a conserved PMSR (peptide methionine sulfoxide reductase) domain and that *MsMsrBs* had a conserved SelR (selenoprotein R) domain (Figure 2). In addition, *MsMsrA1*, *MsMsrA3,* and *MsMsrA4* had another identical domain, namely a low-complexity region, as did *MsMsrA5*, *MsMsrA6,* and *MsMsrA7*, which had the same domain, namely a transmembrane domain. It is worth mentioning that *MsMsrA7* contained not only the conserved domain PMSR and partially shared domain transmembrane domain, but also had peculiar domain of other family, i.e., the rotamase family, and *MsMsrA8* together with *MsMsrA9* only had the single conserved domain PMSR. In MsMsrB, in addition to the conserved SelR domain of the MsrB gene family, *MsMsrB2* contains two low-complexity domains, and *MsMsrB6* contains conserved domains of two other families, namely the Na/H exchanger domain and the TrkA-N domain. *MsMsrB1*, *MsMsrB3*, *MsMsrB4*, and *MsMsrB5* have a single conserved domain SelR. In general, PMSR and SelR are highly conservative in the MsMsrA subgroup and MsMsrB subgroup, respectively.

A phylogenetic tree of the *MsMsrs* in alfalfa showed two obvious subgroups: the MsMsrA subgroup and MsMsrB subgroup. The MsMsrA subgroup contained nine family members from *MsMsrA1* to *9*, and the MsMsrB subgroup contained six family members from *MsMsrB1* to *6* (Figure 3A). The homology of the *MsMsr* was relatively higher among the following: *MsMsrA1*, *2*, and *4*; *MsMsrA3*, *8*, and *9*; *MsMsrA5* to *7*; *MsMsrB1* to *3*; *MsMsrB4* to *6*; additionally, the motif distribution as well as gene structure was more similar (Figure 3B,C). The results of MEME analysis showed that the MsMsr gene family members contained different numbers and types of motifs (Appendix A). There were more motifs in members of the MsMsrA subfamily than in the members of the MsMsrB subfamily, indicating that the two subfamilies had substantial divergence. However, there were no significant differences in the type and number of motifs in each subfamily, indicating that the members of the same subfamily had high similarity. There are six common motifs in the MsMsrA subfamily (motif1, motif2, motif3, motif4, motif6, and motif9). Among them, *MsMsrA1* and *MsMsrA4* have the same composition and order of motifs, while *MsMsrA3*, *MsMsrA8*, and *MsMsrA9* have the same composition and order of motifs, indicating that they are more closely related, which is consistent with the analysis results of the phylogenetic relationship. There are only three common motifs in the MsMsrB subfamily (motif1, motif2, and motif9). These three motifs are also the only three themes in *MsMsrB4*, and the only common themes in the entire MsMsr family. In the MsMsrB subfamily, *MsMsrB1*, *MsMsrB2*, and *MsMsrB3* have the same motif composition and sequence, while *MsMsrB5* and *MsMsrB6* have the same motif composition and sequence and a change in the position of motif1 compared to other *MsMsrBs*, indicating that they are more closely related, which is consistent with the analysis results of the phylogenetic relationship, and demonstrates that the MsMsrA and MsMsrB subgroups are highly conserved, respectively.

A gene structure analysis showed that MsMsr family members had a minimum of 2 exons (*MsMsrA2*, *3*, *8*, and *9*) and 20 exons at most (*MsMsrB6*) (Figure 3C). Four other genes had three exons (*MsMsrA1* and *4*, and *MsMsrB4* and *5*). Two genes had four exons (*MsMsrA6* and *MsMsrB3*), three genes had five exons (*MsMsrA5*, *MsMsrB1* and *2*), and *MsMsrA7* had seven exons. It is worth noting that the CDS length of *MsMsrB6* with 20 exons was also the longest in the family (i.e., 2796 bp). A gene sequence exceeding 2000 bp has not been reported to date in the literature on *Msr* genes, and *MsMsrB6* also had 20 exons. These findings further indicate that even though *Msr* genes have been identified and studied in a variety of plants, the functions of these genes differ among species, emphasizing the importance of functional assays.

### 2.4. Chromosomal Mapping and Collinearity Analysis of MsMsr Genes

MsMsr genes were unevenly distributed across the 12 chromosomes of alfalfa (Figure 4). Three chromosomes contained two MsMsr genes (chr3.4, chr8.1, and chr8.4) and nine chromosomes contained one MsMsr gene (chr2.1, chr3.1, chr3.2, chr4.3, chr4.4, chr5.3, chr5.4, chr8.2, and chr8.3).

Gene duplication is essential for genome and gene system evolution. In alfalfa, 15 MsMsr family members were involved in gene duplication events (Figure 5). Tandem duplications were not detected, and all duplication events were segmental duplications, indicating the expansion mechanism of the MsMsr family in alfalfa and proving that *MsMsrs* are highly conserved in alfalfa. In genetics, *K*_a_/*K*_s_ represents the ratio between rates of nonsynonymous substitutions (*K*_a_) and synonymous substitutions (*K*_s_). Except for *MsMsrA5* and *MsMsrA6* gene pairs, the values of other MsMsr genes pairs were all less than one, consistent with purifying selection (Appendix A). However, *MsMsrA5* and *MsMsrA6* gene pairs showed a ratio greater than one, the signature of positive selection.

### 2.5. Cis-Regulatory Elements in MsMsr Gene Promoters

Cis-regulatory elements are specific DNA sequences located upstream of the gene coding sequence that can bind to regulatory proteins. The promoter regions of the MsMsr gene family were analyzed using PlantCARE^10^ and New PLACE together to predict gene function [37,38]. In addition to the basic TATA box, CAAT box, and light-responsive elements, we mainly analyzed three cis-regulatory elements in the MsMsr genes promoter region: hormone-responsive elements, stress-responsive elements, and plant growth and development elements (Figure 6). It was shown that the promoters of MsMsr genes contained various cis-acting elements with different numbers (Figure 6A and Appendix A). The stress-responsive elements include the wound response element (WUN-motif), defense and stress-responsiveness (TC-rich repeats, STRE, W-boxes), low-temperature and salt stresses (DRE core), low-temperature-responsiveness (LTR), damage and defense response elements (WRE3), and drought-inducibility (MBS) (Figure 6B). The hormone-responsive elements include anaerobic induction (ARE), MeJA-responsiveness (CGTCA-motif, TGACG-motif), ethylene-responsiveness (ERE), salicylic acid-responsiveness (TCA), abscisic acid-responsiveness (ABRE), gibberellin-responsiveness (P-box), flavonoid biosynthetic genes regulation (MBSI), and auxin-responsiveness (TGA, AuxRR-core). Plant growth and development elements include the proliferation-activating response (CCGTCC-box).

The results indicate that anaerobic induction-responsive elements are widely distributed in the promoter regions of the MsMsr gene family, with all 15 members of the family containing such elements. Most of the MsMsr genes also contain four hormone-responsive elements: MeJA-responsiveness, ethylene-responsiveness, salicylic acid-responsiveness, and abscisic acid-responsiveness. Additionally, 13 of the genes have MeJA-responsiveness elements, 11 have ethylene response elements, 12 have salicylic acid response elements, and 13 have abscisic acid response elements, with *MsMsrA7* having 11 abscisic acid response elements. Only *MsMsrA3* contains hormone-responsive elements involved in the regulation of flavonoid biosynthesis. As for stress-responsive elements, the promoter regions of most MsMsr genes contain defense and stress-responsive elements, with some also containing low-temperature, salt stress, damage, and defense response elements, as well as drought-inducibility elements. Specifically, 13 genes have defense and stress-responsive elements, 8 have low-temperature response elements, 7 have drought response elements, and 5 have damage and defense response elements. Only the promoter region of *MsMsrA2* contains low-temperature and salt stress response elements.

### 2.6. Expression Analysis of Homologous Genes of 15 MsMsr Genes under Abiotic Stress

To analyze the expression of MsMsr gene family under salt stress, we assembled and analyzed the transcriptome of alfalfa “Zhongmu No. 1” by referring to the genome of “Xinjiang Daye”. A total of 15 *MsMsr* homologues of “Xinjiang Daye” alfalfa were identified by comparing the transcriptome data treated with “Zhongmu No. 1” alfalfa salt. The expression levels of these 15 genes in salt-treated leaves were analyzed in blank group A (Hoagland nutrient solution) and salt group B (100 mM NaCl+ Hoagland nutrient solution). On this basis, the expression levels of the transcriptome on the first day of blank treatment, the first day of salt treatment, and the sixth day of salt treatment were used for expression analysis. The 15 MsMsr genes were divided into 3 groups based on similarities in the expression patterns of the genes in the transcriptome (Figure 7A). Among them, seven genes showed an upregulated expression trend on the first and sixth days of salt treatment (*MsMsrA1-A3*, *MsMsrA5*, *MsMsrA7*, *MsMsrA8*, and *MsMsrB5*). The expression levels of *MsMsrB2* and *MsMsrB6* increased on the first day and decreased on the sixth day. Three genes showed upregulation trend only on the sixth day (*MsMsrA4*, *MsMsrA9*, and *MsMsrB4*). Three genes showed downregulated expression levels on day 1 and day 6 (*MsMsrA6*, *MsMsrB1*, and *MsMsrB3*).

### 2.7. qRT-PCR Analysis of 15 Homologs of MsMsr Genes 

To verify the results of RNA-Seq and to more specifically analyze the expression of MsMsr genes in alfalfa under salt stress, “Zhongmu No. 1” was treated with 100 mM NaCl on day 25 after germination, and expression levels in alfalfa roots, stems, and leaves were analyzed at 0 h, 3 h, 6 h, 12 h, 24 h, and on day 6 (Appendix A). In addition, this study also treated alfalfa with 300 mM mannitol and 10 μM ABA and analyzed the expression levels of the MsMsr genes in the roots, stems, and leaves of alfalfa at 0 h, 3 h, 6 h, 12 h, 24 h, and the second day for mannitol treatment, and at 0 h, 3 h, 6 h, 12 h, and 24 h for ABA treatment, in order to enrich the response relationship between the MsMsr gene family and abiotic stress.

The 15 homologous genes were all expressed in the roots, stems, and leaves of alfalfa. The expression levels of different genes were different, and the expression levels of the same genes were also different in different tissues (Figure 7B). Among the 15 genes, 7 genes had the highest expression level in leaves (*MsMsrA1*, *MsMsrA4*, *MsMsrA7*, *MsMsrB1*, *MsMsrB2*, *MsMsrB3*, and *MsMsrB6*). The expression levels of six genes (*MsMsrA2*, *MsMsrA3*, *MsMsrA5*, *MsMsrA8*, *MsMsrA9*, and *MsMsrB4*) were highest in the roots. These results suggest that the majority of MsMsr genes have the highest expression levels in both roots and leaves. Among them, *MsMsrB5* is the gene with the highest expression level in roots, and it is also the gene with the highest expression level in stems, indicating that this gene is the gene with a relatively prominent expression level among family members. *MsMsrA1* was expressed at the highest level in leaves. For *MsMsrA3* and *MsMsrA9*, the overall expression levels of the two genes were low in the MsMsr gene family.

qRT-PCR showed that most of MsMsr genes were induced by salt stress (Figure 8 and Appendix A). All 13 MsMsr genes except *MsMsrA6* and *MsMsrA8* showed varying degrees of upregulation in the roots. The expression levels of eight genes in roots under salt stress reached their peak at 3 h, 6 h or 24 h, and the expression levels of five genes reached their peak at 6 d. In addition, all the 14 genes except *MsMsrA8* showed upregulated expression in the stem. The expression levels of 10 genes reached their peak at 6 d, and the expression levels of 4 genes reached their peak at 3 h, 6 h, and 12 h, respectively. These results indicated that the expression of MsMsr genes in the stem of alfalfa was increasing under salt stress, and the expression level changed slowly. In leaves, except for *MsMsrA6*, *MsMsrA8*, and *MsMsrB2*, the expression levels of 12 MsMsr genes were upregulated to varying degrees. Among them, four genes reached the peak expression level at 3 h, three genes reached the peak expression level at 24 h, and five genes reached the peak expression level at 6 d. In particular, salt-induced *MsMsrA7* expression levels in different tissues increased significantly and showed a trend of increasing over time. In addition, the expression levels of *MsMsrB1* and *MsMsrB3* in the roots were significantly upregulated. *MsMsrA2* and *MsMsrA4* were significantly upregulated in stems (*p* < 0.05). It is worth mentioning that the salt-induced MsMsr gene expression in leaves was downregulated and then upregulated. For example, the expression levels of *MsMsrB3* and *MsMsrB4* were downregulated at 3 h, 6 h, and 12 h, and upregulated at 24 h and 6 d. In addition, *MsMsrA1*, *MsMsrA3*, *MsMsrA4*, and *MsMsrA5* were also expressed similarly. In general, the expression levels of 14 MsMsr genes except *MsMsrA8* showed a certain upward trend under salt induction, indicating that the MsMsr gene family mainly showed a positive regulatory relationship under salt stress.

In the promoter analysis of the MsMsr gene family, it was found to contain drought-responsive elements. Therefore, to validate whether the MsMsr gene family responds to drought stress, this study analyzed the expression patterns of MsMsr under drought stress (Appendix A). Under drought stress, most MsMsr genes showed upregulation in expression in the roots, stems, and leaves, and the expression levels peaked mainly within 6–24 h (Figure 9A). Among them, the expression level of *MsMsrA3* in roots, stems, and leaves showed a downward trend within 0–12 h, but an upward trend at 24 h. *MsMsrA7* only showed an upregulation of expression in the leaves, and a downregulation of expression in the roots and stems. *MsMsrB1* and *MsMsrB2* exhibited very similar expression patterns under drought stress, indicating that these two genes are likely to have similar roles in response to drought stress. These results suggest that the MsMsr gene family is likely to respond to drought stress to some extent.

ABA is a plant hormone and one of the key signaling molecules that plants use to respond to stress. Therefore, genes related to the ABA signaling pathway may be closely related to plant stress adaptation, growth, and development. Considering that the promoters of the members of the MsMsr gene family contain abundant ABA response elements, this study also analyzed their expression patterns under ABA stress (Appendix A). Under external ABA treatment, the expression level of stress tolerance genes may be reduced, but in other situations, it may be increased. Under ABA stress, most MsMsr genes showed an overall upregulation trend in roots, stems, and leaves, and the expression level peaked within a short time (3–6 h) (Figure 9B). Unlike most MsMsr genes, the expression level of *MsMsrA3* in roots and stems showed a phenomenon of initial downregulation and subsequent recovery, while the expression level of *MsMsrA7* in roots and stems showed a downward trend, and the expression level of *MsMsrB4* in roots, stems, and leaves showed a downward trend. In addition, the expression levels of *MsMsrB1-3* in roots, stems, and leaves significantly increased during 3–24 h of ABA stress. Most MsMsr genes showed different regulatory patterns in roots, stems, and leaves, which may indicate that the MsMsr gene family responds to ABA stress in different ways, despite different modes of regulation. Based on the above, the MsMsr gene family responds to ABA stress, which is consistent with the results of the promoter element analysis of the MsMsr gene family. This suggests that the MsMsr gene family may play a role in abiotic stress through the ABA-mediated pathway.

## 3. Discussion

In the past, research on alfalfa and abiotic stress has primarily focused on transcription factors, with relatively limited studies on genes, such as oxidoreductases. Although there have been reports on *Msr* in various plants, research on *Msr* in alfalfa has not been conducted yet, indicating that genes, such as *MsMsrs*, hold significant research value and potential.

Several *Msr* genes have been identified in various plants, including Arabidopsis (*n* = 14), soybean (*n* = 12), rice (*n* = 7), and maize (*n* = 8). In this study, 15 MsMsr genes were identified, which is consistent with previous findings where they were divided into two subfamilies and had high conservation. The subcellular localization of the MsMsr gene family proteins was predicted and found to be located in different subcellular structures, including the plastids, cytoplasm, endoplasmic reticulum, mitochondria, and peroxisomes. Five MsMsr proteins were located in the plastids, indicating that these proteins may contribute to maintaining normal physiological functions within the plastids, ensuring the normal growth and development of the entire plant cell. Four MsMsr proteins were located in the cytoplasm, which may play a role in regulating cellular physiological functions, promoting the transport and distribution of substances within the cell, and ensuring normal cell growth, differentiation, and metabolism. Three MsMsr proteins were located in the endoplasmic reticulum, indicating that these three proteins may ensure that different structures and regions within the endoplasmic reticulum can complete their respective biological functions and are also critical to the cell secretion pathway. In addition, the MsMsRB1 and MsMsRB3 proteins were located in the cytoplasm and mitochondria, indicating that these proteins may ensure that different organelle structures and regions can complete their respective biological functions and contribute to regulating cellular metabolism and signal transduction processes. MsMsRB4 was located in the cytoplasm and peroxisomes, indicating that the MsMsrB4 protein may play a role in regulating the activity of metabolic pathways to maintain the normal physiological state of the cell and provide protection against oxidative stress within the cell.

However, it was discovered that in the MsMsr gene family, *MsMsrA7* contained not only the PMSR domain but also the rotamase domain. Similarly, *MsMsrB6* not only contained the SelR domain, but also the Na/H exchanger domain and the TrkA-N domain. In previous studies, there was no occurrence of other conserved domains from gene families in the Msr gene family. This phenomenon may be due to gene fusion. New genes formed by gene fusion can occur between two ancestrally neighboring genes, and sometimes lead to the evolution of novel complex domain structures [39,40]. Previous studies have identified gene fusion events in rice, and these genes may play a role in driving the phenotypic evolution of rice [41]. Rotamase is a type of protein with PPIase activity, and multiple rotamase-encoding genes have been identified in Arabidopsis, some of which are members of the ROF (relative of flavonoid biosynthesis) gene family. In the study of ROF genes, it was found that *ROF1* plays a role in extending thermotolerance by maintaining the levels of small HSPs required for survival at high temperatures [42]. Moreover, *AtROF1* and *AtROF2* in Arabidopsis played a positive role in resistance to acid stress [43]. It is interesting to note that *MsMsrA7* contains both the PMSR conserved domain and the conserved domain of the ROF gene family, which suggests that it may have biological functions similar to those of *ROF* genes. The presence of a large number of ABA response elements in the promoter of *MsMsrA7* further supports the idea that it may play an important role in stress resistance, as ABA is known to be involved in regulating plant responses to abiotic stress. The Na/H exchanger domain and the TrkA-N domain in *MsMsrB6* are not directly connected, and each belongs to a separate gene family. The TrkA-N domain belongs to the NAD-binding component protein, which is commonly used for catalyzing redox reactions. The Na/H exchanger domain belongs to the NHX (Na^+^/H^+^ exchanger) gene family. These proteins balance the ion concentrations inside and outside the cell by transporting H+ from inside the cell to outside and Na^+^ from outside to inside the cell, which plays an important role in plant adaptation to stress. In Arabidopsis, the growth and development of double mutant *atnhx1 atnhx2* are restricted, and the plant is extremely sensitive to external K^+^ and has reduced salt tolerance [44]. Furthermore, previous studies have shown that overexpression of *GhNHX1*, *AtNHX1*, *ScNHX1*, and *TaNHX1* in Arabidopsis enhances the plant’s tolerance to drought stress [45]. In rice, the expression of *HtNHX1* and *HtNHX2* improves the plant’s salt tolerance, and the expression of *HtNHX1* increases rice yield under nutrient-limited conditions [43]. These results suggest that NHX proteins have the characteristics of stress tolerance to abiotic stress, and the conserved domain of the NHX gene family is present in *MsMsrB6*. Moreover, multiple stress-responsive elements are present in the promoter of *MsMsrB6*, which further indicates that *MsMsrB6* is likely a key gene for resistance to abiotic stress. Therefore, further studies are needed to verify the gene function and elucidate the gene regulation mechanism and molecular mechanism of *MsMsrA7* and *MsMsrB6*. Numerous reports have demonstrated the relevance of the *Msr* gene in response to abiotic stress. For example, in maize, *ZmMSRA2* was shown to resist osmotic stress by regulating ROS, proline, and ABA pathways in Arabidopsis, while *ZmMSRA5.1* enhanced plant tolerance to salt stress by reducing ROS accumulation and altering ion transport [46]. Similarly, *TaMsrA3.1*, *TaMsrA4.1*, and *TaMsrA5.2* in wheat were found to improve plant tolerance to salt and drought stress through their catalytic activity and regulatory effects on ROS and ABA pathways [20,21,22]. In rice, the expression of *OsMSRA4.1* was significantly enhanced under salt, drought, low temperature, and high temperature stress, and after ABA treatment. Overexpression of *OsMSRA4.1* reduced ROS levels and enhanced plant tolerance to salt stress [17]. Moreover, in yeast, overexpression of the soybean *GmMSRBs* gene improved yeast tolerance to oxidative stress, and interaction between *GsCBRLK* and *GsMSRB5a* enhanced plant tolerance to alkaline stress [29]. Overexpression of *BpPMSR3* in cabbage also enhanced Arabidopsis tolerance to cadmium [47]. Our findings are consistent with these results, as it can be inferred from the promoter elements of the MsMsr gene family that its members may be involved in various stress responses and hormone pathways, particularly those associated with the widely distributed functional elements within the MsMsr gene family. In the promoter of the MsMsr genes in alfalfa, we identified a large number of stress-responsive elements, such as TC-rich repeats, W-boxes, and MBS, suggesting that it may respond to abiotic stress. Transcriptomics, as a high-throughput analysis technique, can comprehensively measure all genes in cells or tissues, and is widely used in various fields, such as basic research, agriculture, medicine, and environmental science. It has particularly high potential for application in life sciences. For example, previous studies have used transcriptomic techniques to explore the gene expression patterns of Arabidopsis under multiple environmental stress conditions and identified the *RAP2.4* gene related to the immune response to *Botrytis cinerea* [48]. In addition, previous studies have used transcriptome sequencing to reconstruct the transcriptional profiles of organic pollutant degradation and heavy metal stress responses, evaluating the effectiveness of a biotechnology strategy [49]. This study analyzed the expression patterns of the MsMsr gene family under salt stress using transcriptomics and found that the MsMsr gene family may respond to salt stress. Therefore, this confirmed the direction for further research. 

In addition to *MsMsrA8* and *MsMsrA9*, all promoter regions of MsMsr genes contain ABRE, suggesting that MsMsr genes may respond to ABA and regulate plant abiotic stress tolerance via this mechanism. To test this hypothesis, we conducted salt, drought, and ABA stress treatments on alfalfa and found that most MsMsr genes in roots, stems, and leaves were upregulated under salt and drought stress, with a general trend of upregulation under ABA stress. Only *MsMsrA3* and *MsMsrA7* showed downregulation. These results indicate that MsMsr genes may enhance plant resistance to abiotic stress, possibly mediated by the ABA signaling pathway. Nevertheless, further molecular experiments are needed to confirm the role of MsMsr genes in abiotic stress tolerance and to explore their other biological functions.

Overall, the findings of this study on the MsMsr gene family in alfalfa offer valuable insights into the role of *MSRs* in plants, particularly in terms of enhancing resistance to abiotic stress. By identifying two genes potentially involved in abiotic stress tolerance, *MsMsrA7* and *MsMsrB6*, this study offers direction for potential breeding or technological methods to improve the abiotic resistance of crops. This study contributes to the current understanding of plant biology and can serve as a starting point for further research on the role of *MSR* genes in plants.

## 4. Materials and Methods

### 4.1. Identification of MsMsr Genes in Alfalfa

The “Xinjiang Daye” genome was used to identify Msr genes in alfalfa, and is available at https://figshare.com/projects/whole_genome_sequencing_and_assembly_of_Medicago_sativa/66380 (accessed on 1 October 2022). The alfalfa protein sequence and genome annotations were downloaded from the Alfalfa Breeder’s Toolbox (Appendix A). Using alfalfa protein sequences as a reference, the hidden Markov model (HMM) profiles of *MsrA* (PF01625) and *MsrB* (PF01641) were downloaded from the Pfam database (http://pfam.xfam.org) (accessed on 3 October 2022). The Simple HMM Search in TBtools was used for comparisons [50]. With default parameters, the CD-HIT web server (http://www.bioinformatics.org/cd-hit) (accessed on 1 October 2022) was used to delete redundant data. NCBI CD-Search (https://www.ncbi.nlm.nih.gov/Structure/cdd/wrpsb.cgi) (accessed on 4 October 2022) was used for bidirectional authentication. To further analyze the screened family members, the ExPASy proteomics server was used to predict the physicochemical properties of each MsMsr protein, including molecular weight (MW) and theoretical isoelectric point (pI) [51].

### 4.2. Phylogenetic Analysis, Gene Structure, and Motif Composition of MsMsr Genes

For a comprehensive phylogenetic analysis of the *Msr* family, protein sequences of *Msr* genes were downloaded from the Arabidopsis Information Resource (TAIR9) (www.arabidopsis.org) (accessed on 6 October 2022) database. The soybean genome and protein file were downloaded from NCBI (https://ftp.ncbi.nlm.nih.gov/genomes/all/GCF/000/004/515/GCF_000004515.6_Glycine_max_v4.0) (accessed on 5 October 2022). The *Msr* family members of soybean were identified by the same method to identify *Msr* family members in alfalfa. MEGA 7.0 (https://www.megasoftware.net) (accessed on 6 October 2022) was used to build a phylogeny based on the protein sequences of 15 MsMsr genes, 14 *AtMsr* genes, and 22 *GmMsr* genes, a phylogenetic tree was constructed using the maximum likelihood (ML) method. The P-distance model was used with 1000 replications and pairwise detection. All MsMsr genes were divided into the MsMsrA subgroup and MsMsrB subgroup according to their evolutionary relationships with *Msr* genes in Arabidopsis and soybean. The full-length amino acid sequences of MsMsr genes were compared using MEGA 7.0. By using the genome and gene annotation information of alfalfa, and using TBtools, the gene structures of 15 MsMsr genes were visualized and analyzed. MEME (https://meme-suite.org/meme/tools/meme) (accessed on 6 October 2022) was used to predict the MsMsr protein sequences of conserved motifs [52]. TBtools was used to visualize the motif distribution.

### 4.3. Chromosomal Mapping, Gene Duplication, and Collinearity Analysis

Gene Location Visualize from the GTF/GFF function of TBtools was used to visualize the chromosomal locations according to the genome annotation information. In addition, to detect gene duplication events, TBtools was used for a collinearity analysis of 15 MsMsr genes. Based on the results of the collinearity analysis, TBtools was used to calculate the rates of non-synonymous substitutions (*K*_a_) and synonymous substitutions (*K*_s_) for each pair of duplicated genes to analyze selection pressure.

### 4.4. Analysis of Cis-acting Elements in Promoter Regions

Using the genome and annotation information of the “Xinjiang Daye”, 15 upstream 2000 bp promoter sequences of MsMsr genes were extracted via TBtools. In previous studies, the online program PlantCARE^10^ (http://bioinformatics.psb.ugent.be/webtools/plantcare/html) (accessed on 7 October 2022) has been widely used for promoter element analysis [53,54]. Therefore, we chose PlantCARE^10^ along with New PLACE (https://www.dna.affrc.go.jp/PLACE/?action=newplace) (accessed on 8 October 2022) to predict cis-regulatory elements in the upstream 2000 bp region of the MsMsr genes. The obtained results were manually analyzed and sorted. TBtools was used to visualize the promoter elements and draw heat maps of promoter element classification.

### 4.5. Transcriptome and Functional Enrichment Analyses of MsMsr Genes under Salt Treatment

We have previously carried out RNA-Seq analysis of “Zhongmu No. 1” under salt stress in alfalfa. As we used the genome of the “Xinjiang Daye” variety, we used the sequences of 15 MsMsr genes from that variety as templates and used TBtools to perform Blast comparisons on the transcriptome sequences. We identified the corresponding 15 MsMsr genes, and NCBI CD-Search confirmed that all 15 genes belonged to the Msr gene family. We used the Majorbio online (https://www.majorbio.com) (accessed on 1 October 2022) platform to generate a heat map of gene expression in alfalfa under salt stress. The primer sequences used in this study are shown in Appendix A.

### 4.6. Plant Materials and Growth Conditions

The “Zhongmu No. 1” seeds were placed in a petri dish with moist filter paper. After 5 days in the dark, we transferred the germinated seedlings to hydroponic nutrient solution in Hogland’s solution, with the nutrient solution changed every 7 days. The plants were placed in an artificial climate chamber with a light cycle of 16 h of light and 8 h of darkness, a day/night temperature cycle of 25 °C/22 °C, and a relative humidity of 60–70%. After 25 days, alfalfa was subjected to 100 mM NaCl, 300 mM mannitol, and 10 uM ABA stress treatment based on the Hogland’s nutrient solution. Roots, stems, and leaves were collected at 0 h, 3 h, 6 h, 12 h, 24 h, and 6 d after salt treatment, and at 0 h, 3 h, 6 h, 12 h, 24 h, and 48 h after drought treatment, and at 0 h, 3 h, 6 h, 12 h, and 24 h after ABA treatment. All samples were quickly frozen in liquid nitrogen and stored at −80 °C until use.

### 4.7. Quantitative Real-Time PCR Analysis

Fifteen MsMsr genes were selected for qRT-PCR analysis to validate the transcriptome data of alfalfa under salt stress and to enrich the real-time expression data of the MsMsr gene family under salt, drought, and ABA stress [55]. We extracted RNA using the FastPure Plant Total RNA Isolation Kit (Vazyme Biotech Co., Ltd., Nanjing, China) following the manufacturer’s instructions. We used Thermo Scientific™ NanoDrop™ One Microvolume UV-Vis Spectrophotometers to quantify RNA samples, and then assessed RNA integrity through 1% agarose gel electrophoresis. Specifically, we used a 1% agarose gel to visualize RNA bands under UV light. The RNA was reverse-transcribed into cDNA using the HiScript III^®^ RT SuperMix for qRT-PCR (+gDNAwiper) (Vazyme Biotech Co., Ltd., Nanjing, China), with consistent input amounts of RNA. We performed qRT-PCR using ChamQ SYBR Color qRT-PCR MasterMix (Vazyme Biotech Co., Ltd., Nanjing, China) in the Bio-Rad CFX96 Touch System, following the manufacturer’s instructions. The qRT-PCR program consisted of a 30 s reaction at 95 °C, followed by 40 cycles of PCR, with each cycle consisting of a 10 s reaction at 95 °C and a 30 s reaction at 60 °C. The *MsActin* gene (AA660796) was used as the internal reference gene in qRT-PCR [56,57]. This sentence describes a qRT-PCR analysis of the MsMsr gene family under salt stress. Using the Bio-Rad CFX Maestro software 1.0 in conjunction with the LinRegPCR program, the amplification efficiency can be detssermined. The experiment was conducted using biological triplicate samples, and the data were analyzed using the 2^−∆∆CT^ method. The results of the qRT-PCR analysis were presented as the mean ± standard deviation (SD) of the expression levels at each time point under salt stress and were then compared to the expression levels at 0 h. Statistical significance was determined using Delta Cq data, and the significance is calculated using Student’s *t*-test (*n* = 3, * *p*< 0.05). The t-test was used to determine the statistical significance of the expression levels of the three replicates and their standard deviation under salt stress. Heatmaps were generated to present the expression levels at each time point under drought and ABA stress, with no clustering of the horizontal and vertical coordinates.

## 5. Conclusions

This study conducted a systematic and comprehensive bioinformatics analysis of the MsMsr gene family in alfalfa for the first time. The expression patterns of the MsMsr genes under salt, drought, and ABA stress were analyzed using transcriptome data and qRT-PCR technology, enriching our understanding of this gene family and predicting its potential biological functions in alfalfa.

This study identified 15 MsMsr genes in the “Xinjiang Daye” alfalfa genome. To investigate the functions of the MsMsr gene family members, we predicted the subcellular localization of the 15 MsMsr proteins. Our results showed that they were primarily located in the endoplasmic reticulum and its organelles, suggesting diverse functions. In addition, to study the evolutionary relationship of the MsMsr gene family, we constructed phylogenetic trees of multiple and single species and analyzed the MsMsr genes sequences. Our findings revealed that the MsMsr gene family was divided into two highly conserved subfamilies, MsMsrA and MsMsrB. Notably, *MsMsrA7* and *MsMsrB6* contained conserved domains from other gene families in addition to the PMSR and SelR domains of the MsrA and MsrB subfamilies, respectively. In the analysis of promoter elements, the study focused on stress and hormone-related response elements. The results showed that the MsMsr gene family contains a variety of hormone-related response elements, such as ABA response elements and MeJA response elements. Additionally, the gene family contains various stress-related response elements, such as drought-inducibility, low-temperature-responsiveness, and defense and stress-responsiveness elements. These findings indicate that the MsMsr gene family may have a certain response to non-biological stresses. To investigate the relationship between the MsMsr gene family and abiotic stress further, this study used transcriptome data of alfalfa under salt stress. Furthermore, the study validated the findings using qRT-PCR of alfalfa under salt, drought, and ABA stress. The results demonstrate that the MsMsr gene family responds to salt, drought, and ABA stress.

In summary, this study provides a theoretical basis for exploring the gene function and molecular mechanism of MsMsr gene family members, predicts that the MsMsr gene family may play a key role in responding to abiotic stress, and suggests that further research is needed to investigate whether the MsMsr genes have biological functions related to stress resistance and whether it can be used for genetic breeding.

## Figures and Tables

**Figure 1 ijms-24-09638-f001:**
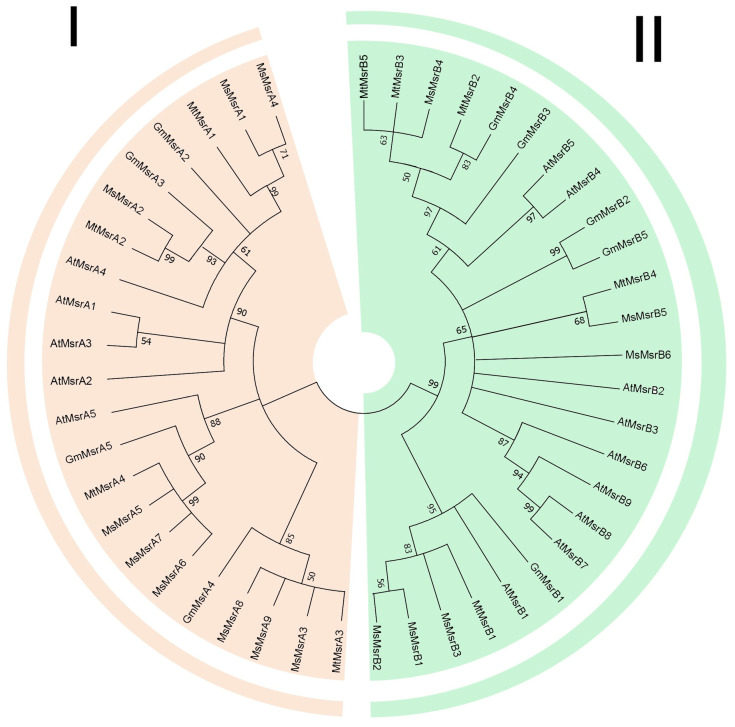
Phylogenetic analysis of *Msr* genes in alfalfa, Arabidopsis, and soybean. I represents the MsrA subfamily and II represents the MsrB subfamily.

**Figure 2 ijms-24-09638-f002:**
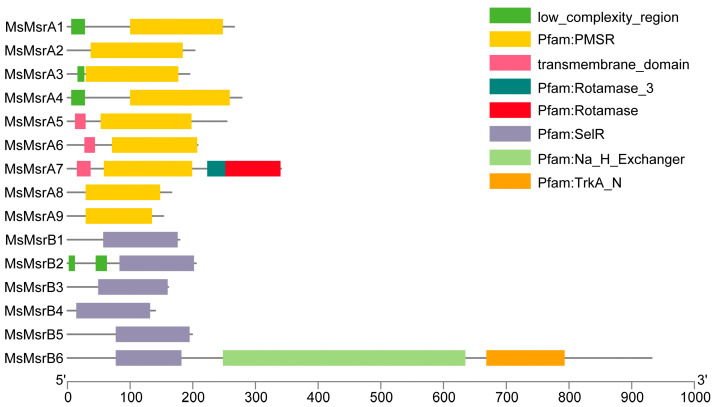
Domain analysis of the MsMsr gene family in alfalfa. Different colored areas represent different domains. The yellow region is the conserved domain of the MsMsrAs. The purple region is the conserved domain of the MsMsrBs. The horizontal coordinate indicates the length of the protein sequence.

**Figure 3 ijms-24-09638-f003:**
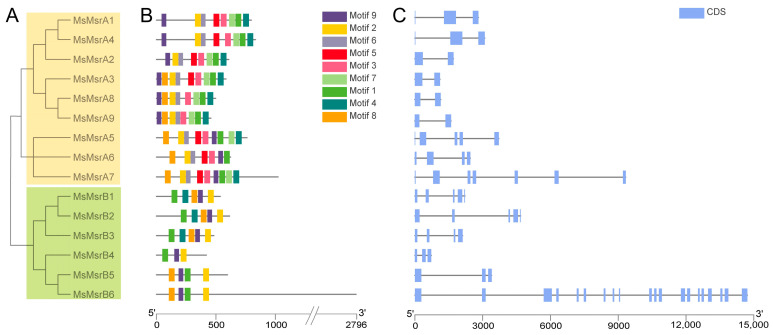
Phylogenetic relationship, motifs, and gene structure analysis of Msr genes in alfalfa. (**A**) Phylogenetic trees of 15 MsMsr genes in alfalfa. (**B**) Conservative motif arrangement of MsMsr genes. Different colored boxes represent different motifs. The horizontal coordinate represents the CDS length of the gene. (**C**) The exon–intron organization of the MsMsr genes. The blue box represents the exon; the black line represents the intron. The horizontal axis represents the full length of the gene.

**Figure 4 ijms-24-09638-f004:**
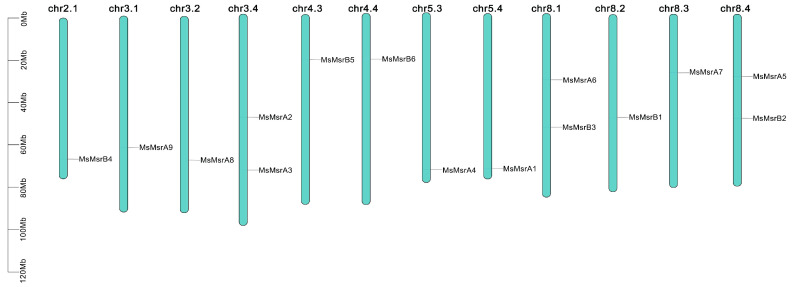
The distribution of MsMsr genes on alfalfa chromosomes. The green bars represent each chromosome, and the black lines label the position of each MsMsr gene. The ordinate is the length of the chromosome.

**Figure 5 ijms-24-09638-f005:**
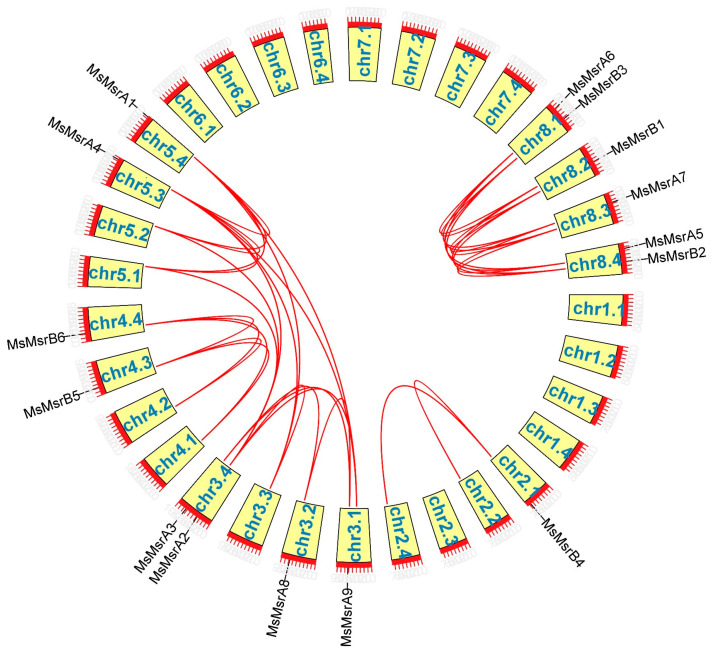
Collinearity analysis of MsMsr genes. The red line shows the MsMsr gene pairs replicated in alfalfa. The yellow box shows the chromosomal position.

**Figure 6 ijms-24-09638-f006:**
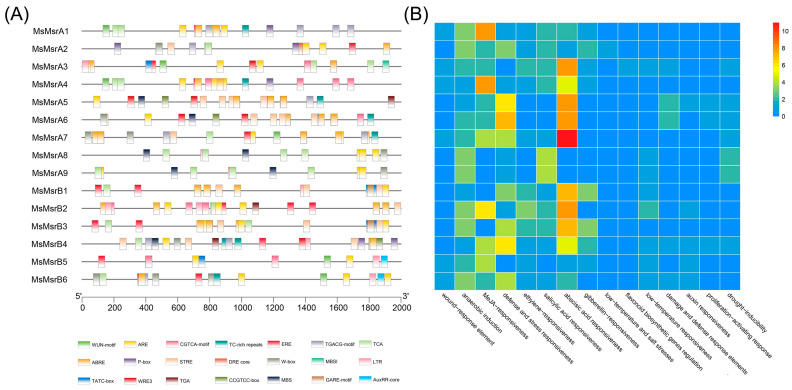
Analysis of the 2000 bp promoter regulatory elements of the MsMsr genes. (**A**) Colored blocks represent different types of cis-regulatory elements and their relative positions within each MsMsr gene promoter region. (**B**) Colors indicate the number of different cis-regulatory elements present in these MsMsr genes, presented in the form of a heatmap.

**Figure 7 ijms-24-09638-f007:**
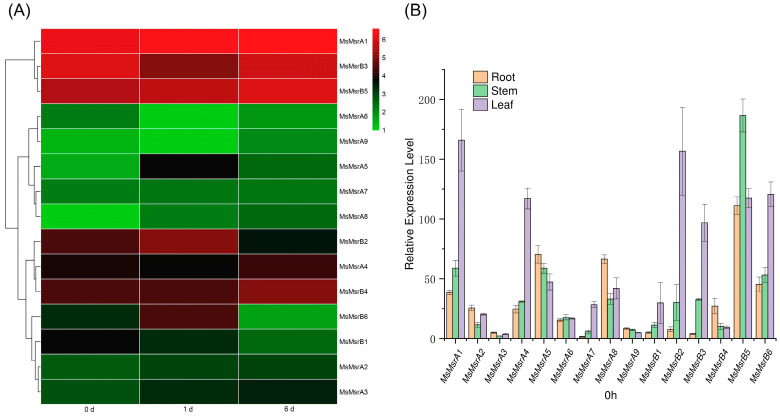
Analysis of expression patterns of 15 MsMsr genes. (**A**) Expression profiles with the log2 (FPKM) values of MsMsr genes in different tissues under salt stress retrieved from transcriptome data. Here, 0 d represents before treatment, 1 d represents the first day of salt treatment, and 6 d represents the sixth day of salt treatment. Red indicates high expression levels, while green indicates low expression levels. (**B**) qRT-PCR analysis of the expression levels of 15 MsMsr genes in roots, stems, and leaves at 0 h. Data are presented as mean ± SD.

**Figure 8 ijms-24-09638-f008:**
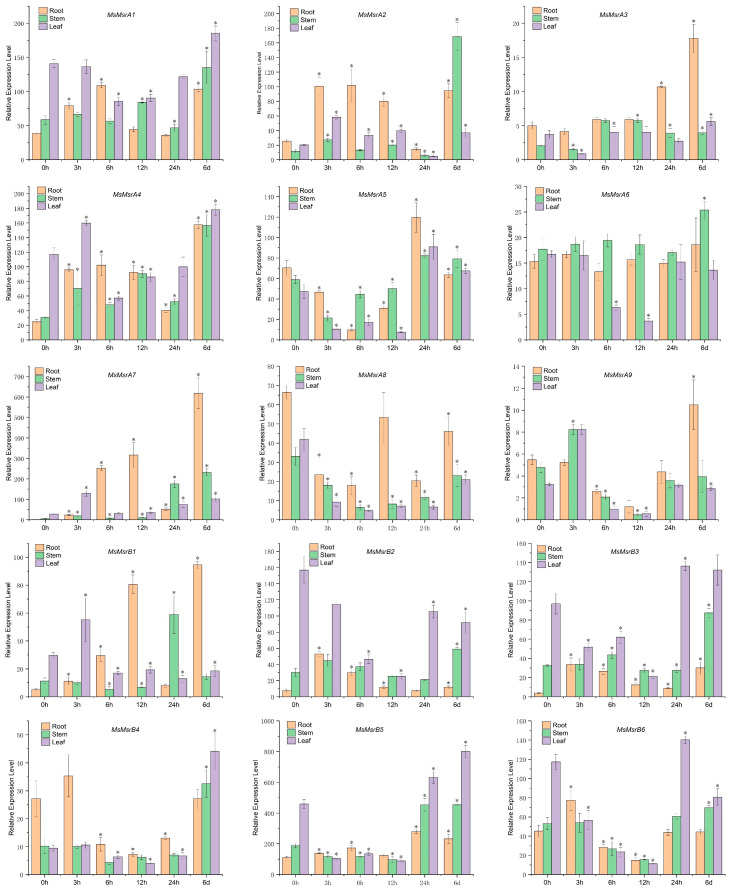
Analysis of expression patterns of 15 MsMsr genes in roots, stems, and leaves under salt stress. Data are presented as mean ± SD, Student’s *t*-test (*n* = 3, * *p* < 0.05).

**Figure 9 ijms-24-09638-f009:**
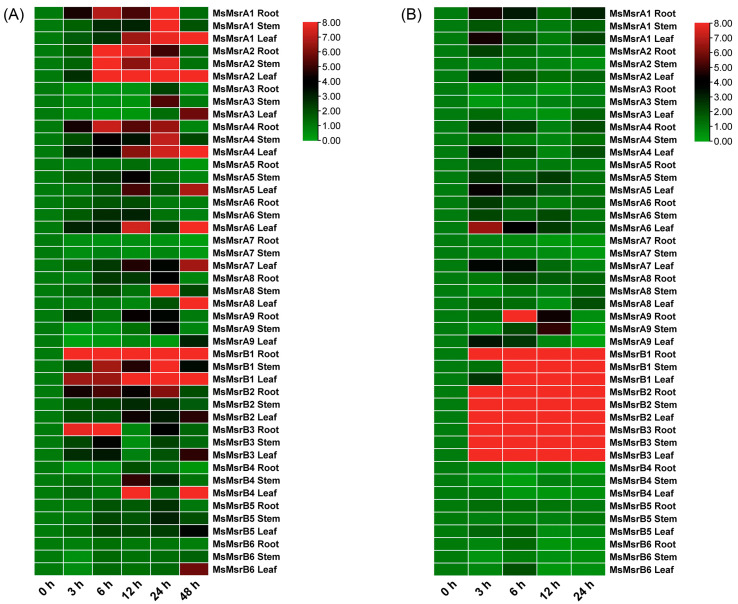
Analysis of expression patterns (fold change, not log-transformed) of 15 MsMsr genes in roots, stems, and leaves under drought and ABA stress. (**A**) Expression profiles of MsMsr genes under drought treatments at different time points. (**B**) Expression profiles of MsMsr genes under ABA treatments at different time points. The horizontal axis represents the time of stress treatment, and the vertical axis represents genes and tissue parts. The results are presented as relative expression levels. Red indicates high expression levels, and green indicates low expression levels.

**Table 1 ijms-24-09638-t001:** Analysis of the physicochemical properties and subcellular localization prediction results of the MsMsr gene family.

Name	Gene ID	CDS (bp)	Length (aa)	pI	MW (Da)	Predicted Location (s)	Predicted Signal (s)
MsMsrA1	MS.gene038415	798	265	8.75	29,419.08	Plastid	Chloroplast transit peptide
MsMsrA2	MS.gene027087	609	202	8.20	23,205.85	Cytoplasm	None
MsMsrA3	MS.gene29083	585	194	5.48	21,737.02	Cytoplasm	None
MsMsrA4	MS.gene017180	834	277	7.53	30,789.65	Plastid	Chloroplast transit peptide
MsMsrA5	MS.gene012096	762	253	5.79	28,908.68	Endoplasmic reticulum	Signal peptide
MsMsrA6	MS.gene44259	624	207	6.42	23,033.94	Endoplasmic reticulum	Signal peptide
MsMsrA7	MS.gene89646	1023	340	6.05	38,010.76	Endoplasmic reticulum	Signal peptide
MsMsrA8	MS.gene022411	498	165	5.34	18,248.1	Cytoplasm	None
MsMsrA9	MS.gene78714	459	152	5.2	16,851.51	Cytoplasm	None
MsMsrB1	MS.gene032356	537	178	8.48	20,193.22	Cytoplasm, mitochondrion	Mitochondrial transit peptide
MsMsrB2	MS.gene38969	615	204	9.09	22,729.57	Plastid	Chloroplast transit peptide
MsMsrB3	MS.gene007821	483	160	6.29	18,119.74	Cytoplasm, mitochondrion	Mitochondrial transit peptide
MsMsrB4	MS.gene002820	420	139	6.08	15,101.88	Cytoplasm, peroxisome	Peroxisomal targeting signal
MsMsrB5	MS.gene40100	597	198	8.87	21,424.38	Plastid	Chloroplast transit peptide
MsMsrB6	MS.gene053626	2796	931	5.7	101,351.14	Plastid	None

## Data Availability

All data in the present study are available in the public database, as referred to in the Materials and Methods section.

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
