# Peer review of "Genome-Wide Identification and Characterization of the Msr Gene Family in Alfalfa under Abiotic Stress"

_ijms, 2023, doi:10.3390/ijms24119638_

Round 1

Reviewer 1 Report

This study consists in a detailed bioinformatic analysis of the Msr gene family in Medicago sativa L. and the study of the gene expression patterns under the influence of abiotic stresses. The manuscript is very interesting and clearly written. However, a few adjustments are necessary.

Page 3: there are discrepancies between the text and the data showed in the table 1.

Lines 103 and 107: please, write the correct names of the MsMsr proteins.

Lines 106-107: "Three MsMsr proteins (MsMsRA5, MsMsRA6 and MsMsRA7) were located in the plastids" what is their correct localisation? (check the table 1).

It might be better to also include MsMsB6 and MsMsB6 in the text.

Correct the errors on lines 160-183-185.

Lines 380-383: this sentence is unclear.

Materials and methods: how was the RNA quantified and which gel was used to assess its integrity?

It would be better to include a brief focus on the relevance of transcriptomics as a leading technique and its potential in various research fields. Take a look at these papers:

Ÿ  Tartaglia, M., Sciarrillo, R., Zuzolo, D., Postiglione, A., Prigioniero, A., Scarano, P., . . . Guarino, C. (2022). Exploring an enhanced rhizospheric phenomenon for pluricontaminated soil remediation: Insights from tripartite metatranscriptome analyses. Journal of Hazardous Materials, 428 doi:10.1016/j.jhazmat.2022.128246

Ÿ  Sham, A., Al-Ashram, H., Whitley, K., Iratni, R., El-Tarabily, K. A., & AbuQamar, S. F. (2019). Metatranscriptomic analysis of multiple environmental stresses identifies RAP2.4 gene associated with arabidopsis immunity to botrytis cinerea. Scientific Reports, 9(1) doi:10.1038/s41598-019-53694-1

Ÿ  Huang, Y., Yi, J., Li, X., & Li, F. (2024). Transcriptomics and physiological analyses reveal that sulfur alleviates mercury toxicity in rice (oryza sativa L.). Journal of Environmental Sciences (China), 135, 10-25. doi:10.1016/j.jes.2023.01.001

Ÿ  Cao, S., Wang, M., Pan, J., Luo, D., Mubeen, S., Wang, C., . . . Chen, P. (2024). Physiological, transcriptome and gene functional analysis provide novel sights into cadmium accumulation and tolerance mechanisms in kenaf. Journal of Environmental Sciences (China), 137, 500-514. doi:10.1016/j.jes.2023.03.006

Ÿ  Kumar, A., Thomas, J., Gill, N., Dwiningsih, Y., Ruiz, C., Famoso, A., & Pereira, A. (2023). Molecular mapping and characterization of QTLs for grain quality traits in a RIL population of US rice under high nighttime temperature stress. Scientific Reports, 13(1) doi:10.1038/s41598-023-31399-w

the quality of the text is good, I would recommend re-reading for typing errors and small imperfections

Author Response

We would like to thank you very much for your positive and constructive comments and suggestions on our manuscript, which we feel have significantly improved the quality of the manuscript. We have carefully revised the manuscript accordingly (the revised text is highlighted by track changes). With this revision, we hope that the manuscript is now acceptable for publication in International Journal of Molecular Sciences.Please see the attachment for responses.

Reviewer 2 Report

This work presents " Genome-wide identification and characterization of the Msr gene family in alfalfa under abiotic stress". This manuscript is recommended to be published after including and addressing the below listed comments with major corrections.

- The authors should eliminate the current grammatical and punctuation mark errors and also confirm the correct scientific English.
- The authors should write the complete terms of all abbreviations (including the instruments) before the first use in the abstract and main manuscript.
- The authors should clearly explain the innovation and importance of their work on the introduction of the manuscript. They should justify the value of the work and compare their work with previously similar published papers. They should develop the advantage and applications of this procedure. The introduction section needs to be elaborated.
- The authors should work on the scientific English of the manuscript and elaborate it.
- The authors should cite important references

- The authors should include the brand and model of all instruments they used in this project.

- The quality of the figure should be improved. 

- The authors should work on the scientific English of the manuscript and elaborate it.

Author Response

We would like to thank you very much for your positive and constructive comments and suggestions on our manuscript, which we feel have significantly improved the quality of the manuscript. We have carefully revised the manuscript accordingly (the revised text is highlighted by track changes). With this revision, we hope that the manuscript is now acceptable for publication in International Journal of Molecular Sciences. Please see the attachment for responses.

Reviewer 3 Report

Review

The manuscript “Genome-wide identification and characterization of the Msr gene family in alfalfa under abiotic stress” by Zhao et al describes the characterization of 15 Msr (methionine sulfoxide reductase) genes. Reading the results, I often missed brief explanations of how the various characterizations were done. It would be helpful to briefly mention the tools used and any information that helps the reader to gauge the significance of the presented data. For example, in the presentation of a phylogenetic tree, the tool (e.g. MegaX), type of tree (e.g. Maximum likelihood), and what type of statistical evaluation (e.g. bootstrap values) should be mentioned.

On the hand, the speculation and interpretation should be reduced in the results, those belong in the discussion.

I had many questions in regard to the presented results (outlined below), that it is difficult to gauge at this point if the data is a significant contribution. Much depends on the expression analysis, which at this point is not clear to me, in part because even the unrtreated controls showed up and down-regulation. I believe the manuscript requires extensive clarification before a decision can be made.

Specific points

Introduction

“Extreme conditions, such as high salinity, drought, and low temperatures, have reduced the yield of alfalfa by at least 10–20% [3].”

Globally, temperature are rising; how do you explain a trend of cold temperatures involved in yield losses?

“Msr gene almost plays different roles in different species, indicating that Msr gene family has great research potential and value.”

What do you mean with “almost”? “Almost” is very vague.

Is there evidence for different functions of orthologs across species, or is just a matter of diverse functions within large gene families? Clarify the text.

Results

Identification of MsMsr family members: this section includes too much vague speculation on function based on location. Just present the results; move interpretations of the results to discussion; avoid any “empty” speculations such as “thereby ensuring the normal growth and development of the entire plant cell”.

“Three MsMsr proteins (MsMsRA5, MsMsRA6, and MsMsRB7) were located in the plastids, indicating that these three proteins may ensure that different structures and regions within the endoplasmic reticulum can complete their respective biological functions, and are also key to the cell secretion pathway.”

Did you mean to say that these were located in the ER? Again, avoid empty speculation; the sentence “indicating that these three proteins may ensure that different structures and regions within the endoplasmic reticulum can complete their respective biological functions, and are also key to the cell secretion pathway.” Is so vague that it really adds nothing. Rather discuss locations of MsMsr proteins in the context of what is known about locations and functions of Msr proteins in other species.

“Among them, 3 MsMsr proteins were predicted to have a Chloroplast transit peptide, indicating that they may have a guiding role in the localization and transport of chloroplast proteins”

This sentence make it sound  like the 3 Msr proteins have a “a guiding role in the localization and transport of chloroplast proteins” which I don’t think you mean. I believe that readers understand that chloroplast transit peptides indicate plastid location, so that information could just be added to the earlier information on “three MsMsr proteins (MsMsRA1, MsMsRA4, and MsMsRB2) were located in the plastids, [supported by the presence of chloroplast transit peptides].” 

2.2. Phylogenetic analysis of Msrs

“Four plants including M. sativa, M. truncatula, G. max and A. thaliana were selected to…”

Replace “plants” with “species”

2.3. Sequence and structure analysis of MsMsrs

“A domain analysis (what tool?) showed that clearly MsMsrAs had has a conserved PMSR domain and MsMsrBs had has a conserved SelR domain.”

Spell out what PMSR and SelR stand for at first mentioning.

2.5. Cis-regulatory elements in MsMsr gene promoters

“Cis-regulatory elements are specific DNA sequence located upstream of the gene coding sequence that can bind to regulatory proteins. We analyzed 15 MsMsrs promoter”

Mention briefly HOW you analyzed promoters, what tools did you use, what is the statistical significance of your findings? While the technical details belong of course in the Methods section, the reader still needs some information in the results to gauge the presented data.

Figure 7.: What are the units of the legend (red, green, 1-6?). Is it log2FC? But in that case, I would expect negative values for down-regulated genes. 

Why is the control (t0) also up and down-regulated? If green and red show fold change, that would be in comparison to the t0. Or is it showing level of expression, e.g. as FPKM? Labeling the legend would help.

2.7. qRT-PCR analysis of 15 homologs of MsMsr genes

“To verify the results of RNA-Seq and more specifically analyze the expression of MsMsr genes in alfalfa under salt stress, “Zhongmu No. 1” was treated with 100 mM NaCl on day 25 after germination, and expression levels in alfalfa roots, stems, and leaves were analyzed at 0 h, 3 h, 6 h, 12 h, 24 h, and on day 6 (Table S4).”

This section is repeated

Figure 9.

Here red and green are clearly fold change (I assume log2FC, label accordingly), but in that case, I do not understand the foldchange in t0, isn’t that your comparison? What does it mean that most genes are differentially expressed at t0 before treatment even started? And compared to what?

Discussion

“This phenomen due to gene fusion.”

This phenomenon may be due to gene fusion.

“By identifying two potential genes with stress tolerance, MsMsrA7 and MsMsrB6,”

By identifying two genes potentially involved in abiotic stress tolerance…

Materials and Methods

“By using the genome and gene annotation information of alfalfa, and using TBtools, the gene structure of 15 MsMsr genes was visualized and analyzed.” Citation needed at first mentioning of TBtools.

Analysis of cis-acting elements in promoter regions

Add information on statistical significance cut off; how was decided which motifs to include?

The MsActin gene was used as the internal reference gene in qRT-PCR.

“Did you test (e.g. with geNorm) if actin expression is stable under the various conditions used?

Was amplification efficiency tested and taken into account when calculating fold change?

“Statistical significance was calculated using Student's t-test” Mention what data you used for t-test, e.g. linear fold-change?

Author Response

(The authors gave the same response as above.)

Round 2

Reviewer 3 Report

My main issue with this manuscript is that it does not seem to add much to the scientific knowledge base. A gene family that has been characterized before in other species now is characterized  in Alfalfa. The paper reads more like a catalogue than an exciting exploration. There is no attempt of making correlations between findings, e.g. between promoter elements and gene expression. The selection of promoter motifs was not statistically evaluated in any way. Such motifs are typically short and frequent; without any statistical evaluation, such a list of possible motifs seems rather meaningless.

Some of the review responses were directed to me, but not changed in the manuscript. For example, there is no mentioning that amplification efficiency was determined and taken into account in the qRT-PCR methods section. And the sentence  “The t-test was used to determine the statistical significance of the expression levels of the three replicates and their standard deviation under salt stress.” still does not explain what data type was used for the ttest. Fold-change? Log2FC? Delta Cq?

My biggest issue with the current manuscript is that I still don’t understand the heatmaps.

There are two main ways to show expression results in heatmaps, as expression level or as fold change. 

If expression level is shown, e.g. as normalized read count in case of RNA-seq, the control is included in a heatmap. In the figure legend and review response, authors indicate that they indeed show the level of expression. However, they also changed the color legend to fold-change, which is a different way to show expression data.

If fold-change is used, the expression during treatment is divided by the expression of the control, in this case by t0 (and of course log2-transformed which I just ignore for now). Authors indicate that they indeed used fold change, compared to t0. But if that is the case, the control should NOT show any fold change, because it would be divided by itself.

So my question remains: if the heatmaps show foldchange, why is the control showing differential expression? And what would explain differential expression in t0, before the treatment even starts? 

I know authors have tried to explain this to me in their response, but I still can’t sort it out. 

Without understanding the heatmaps, I can’t gauge the significance of the findings. 

Overall, English language is fine.

Author Response

Thanks again for your valuable comments. We have revised the manuscript according to your suggestions and realized the problem.

Round 3

Reviewer 3 Report

My major concerns have been addressed.

For figure 9, the legend should state what is meant by "expression pattern". From the author's response, I believe it is fold change (not log-transformed), but that should be stated.

fine

Author Response

Thanks once again for your valuable feedback on the manuscript. We sincerely appreciate it!
